# COVID-19 Alcoholic Cirrhosis and Non-Alcoholic Steatohepatitis Cirrhosis Outcomes among Hospitalized Patients in the United States: Insight from National Inpatient Sample Database

**DOI:** 10.3390/tropicalmed7120421

**Published:** 2022-12-07

**Authors:** Devika Kapuria, Karthik Gangu, Prabal Chourasia, Aniesh Boba, Anthony Nguyen, Moon Ryu, Mark Peicher, Mario Flores, Harleen Kaur Chela, Ebubekir S. Daglilar, Abu Baker Sheikh, Rahul Shekhar

**Affiliations:** 1Division of Gastroenterology, Washington University, St. Louis, MO 63130, USA; 2Department of Internal Medicine, University of Kansas Medical Center, Kansas City, KS 66160, USA; 3Department of Hospital Medicine, Mary Washington Hospital, Fredericksburg, VA 22401, USA; 4Department of Medicine, John H Stronger Hospital, Cook County, Chicago, IL 60612, USA; 5Division of Internal Medicine, University of New Mexico Health Sciences Center, Albuquerque, NM 87106, USA; 6Department of Internal Medicine, Division of Gastroenterology, Charleston, WV 26506, USA

**Keywords:** COVID-19, cirrhosis, alcohol cirrhosis, NASH cirrhosis, mortality, national inpatient sample, United States

## Abstract

Patients with co-morbidities like cirrhosis are at risk of worse outcome from COVID-19 infection. Given limited prior studies, we evaluated outcomes associated with COVID-19 infection in alcoholic and non-alcoholic steatohepatitis cirrhotic (CC+) versus cirrhotic without COVID-19 (CC−). We performed retrospective analysis of 822,604 patients including 28,610 COVID-19 patients from the National Inpatient Sample database with alcoholic and NASH cirrhosis enrolled between 1 January 2020 to 31 December 2020, with univariate and multivariate regression analyses. Primary outcome was mortality and secondary outcomes was mechanical ventilation, vasopressor use, length of stay, hospitalization expense and predictors of mortality. In-hospital mortality was three time higher in the CC+ group compared to those in the CC− group(18.6% vs. 5.96%, *p* < 0.001, adjusted odds ratio (OR)3.39 (95% 3.08–3.74 CI). Hospitalization was more likely for underrepresented racial and ethnic groups with COVID-19 and cirrhosis. CC+ group had over twice the rates of mechanical ventilation (19.92% vs. 9.07%, adjusted OR 2.71 2.71 (95% 2.51–2.93 CI)),1.7 times likelihood of receiving vasopressors (4.12% vs. 2.45%, *p* < 0.001, adjusted OR 1.71 (95% CI 1.46–2.01). COVID-19 is associated with increased mortality in patients with alcoholic and NASH cirrhosis, and patients with alcoholic cirrhosis and COVID-19 have a slightly higher mortality compared to NASH cirrhosis.

## 1. Introduction

COVID-19 affect several organ systems, the most common affected being the pulmonary system; however, high risk co-morbid conditions greatly increase the death and morbidity associated with COVID-19 [1]. Since its emergence, the hepatic involvement by this virus has been demonstrated in several studies and the degree of impact varies significantly [2,3,4,5]. The global burden of patients with chronic liver disease is large, with almost 2 million deaths occurring per year due to hepatic decompensation events and hepatocellular carcinoma [2]. Immune dysregulation is a well-recognized phenomenon in cirrhosis [3], thus making them more susceptible to complications of COVID-19. Patients with cirrhosis are shown to have a poor prognosis with COVID-19 [6,7]. Several studies have reported outcome comparisons among patients hospitalised with cirrhosis and COVID-19 versus patients with cirrhosis without COVID-19. In a multicentre study, Bajaj et al. demonstrated that inpatient mortality was similar in patients with cirrhosis and COVID-19 compared to cirrhotic without COVID-19, however, was higher than a cohort of patients without cirrhosis admitted with COVID-19 infection [8]. Additionally, registry data has shown high mortality rates up to 40% for COVID-19 infection in patients with pre-existing liver disease and cirrhosis [9], which was much higher than that reported in hospitalised cirrhotic patients in the pre-COVID era as well cirrhotic patients admitted with influenza [10,11]. Other studies, however, have shown a much lower incidence of mortality, with a centralised electronic health record data resource reporting a 30 day mortality of 8.9% in patients with cirrhosis with COVID-19 infection, and the presence of COVID-19 in cirrhotic patients being associated with 2.38 times hazard of death [12]. 

There have been few studies characterising the outcomes of patients with cirrhosis and COVID-19 based on the etiology of cirrhosis. In a single-center study of patients with chronic liver disease of mostly alcoholic etiology, hospitalised for COVID-19, presence of cirrhosis was associated with fourfold increase in 30 day mortality compared to patients without cirrhosis [13]. Yoo et al. showed that patients with pre-existing NAFLD have a higher likelihood of testing positive for COVID-19, and are more predisposed to severe illness [14].Additionally, Kim et al. showed that the etiology of all-cause mortality from alcoholic liver disease and non-alcoholic fatty liver disease increased significantly during the COVID-19 pandemic, with increase in cirrhosis-related mortality mainly attributable to alcoholic liver disease [15].

We hypothesize significant morbidity and mortality in patients with cirrhosis and COVID-19 infections, with respiratory compromise as the driving force for decompensatory episodes. The aim of this study was is to evaluate outcomes associated with inpatient hospitalisation of patients with alcoholic and NASH cirrhosis and COVID-19, and to assess disease severity and outcomes based on etiology of cirrhosis, namely alcoholic cirrhosis and non-alcoholic fatty liver disease. 

## 2. Materials and Method

### 2.1. Data Source

This retrospective study utilized the Agency for Healthcare Research and Quality (AHRQ) 2020 NIS dataset, which is based on hospitalizations from 1 January 2020 to 31 December 2020. The NIS is a nationwide administrative database developed by the Agency for Health Care Research and Quality (AHRQ) as part of the Health Care Cost and Utilization Program (HCUP). Approximately 20% of all United States (U.S.) hospitalizations are represented within this database. These hospitalizations are a nationally representative sample of all-payer discharges. All patients who were 18 years of age and older and were admitted to the hospital with history of liver cirrhosis were included in the study. This group was further divided based on concurrent diagnosis with COVID-19.

International classification of diseases 10th—clinical modification (ICD-10-CM) codes were used to retrieve patient samples with comorbid conditions and ICD-10 procedure codes were used to identify inpatient procedures. A detailed code summary is provided in Appendix A.

### 2.2. Covariates

NIS data sample contains data regarding in-hospital outcomes, procedures, and other discharge-related information. Variables were divided into patient level, hospital level, and illness severity.

a.Patient-level: Age, race, sex, comorbidities, insurance status, income in patient’s zip code, disposition.b.Hospital level: Location, teaching status, bed size, region.c.Illness severity: Length of stay (LOS), mortality, hospitalization cost, Elixhauser comorbidity score, in-hospital complications, mechanical ventilation, circulatory support and vasopressor use, NACSELD-ALF score.

### 2.3. Study Outcomes

Primary outcome assessed was in-hospital mortality. Secondary outcomes were (a) intubation rate and vasopressor use, (b) length of stay, (c) financial burden on healthcare, (d) disposition & resource utilization, and (e) in-hospital outcomes of COVID positive alcoholic cirrhosis compared to COVID positive NASH cirrhosis.

### 2.4. Statistical Methods

STATA 17 (StataCorp LLC, College Station, TX, USA) was utilized for statistical analysis. Unweighted sample was 6.47 million observations and weighted sample was around 32.3 million discharges for year 2020. Patients who were admitted with history of cirrhosis were retrieved with ICD-10 CM codes and this group was further divided based on COVID-19 status. Chi-square test was used to compare categorical variables and linear regression was used for continuous variables. For primary outcome univariate logistic regression was used to calculate unadjusted odds ratio for variables of interest and *p* values of ≤0.2 on univariate logistic regression was used to build multivariate logistic regression model to adjust for potential confounders.

Multivariate linear regression model was used for continuous variables (LOS and Total hospital charge) [16]. A two-tailed *p*-value of 0.05 was considered significant. Subgroup analysis between COVID positive alcoholic cirrhosis and COVID positive NASH cirrhosis was performed with the method described above.

Predictors of mortality in COVID positive cirrhotic patients were calculated using multivariate cox regression analysis and this model was built using the method described above for multivariate logistic regression analysis.

## 3. Results

Between 1 January 2020 and 31 December 2020, there were 822,604 patients hospitalised with a clinical diagnosis of either alcoholic or NASH cirrhosis. Of these, 28,610 patients were positive for COVID-19.

### 3.1. Patient Characteristics 

Forty percent of the cohort was female. There was no significant difference in sex distribution between the COVID-19 positive alcoholic and NASH cirrhosis (CC+) and COVID-19 negative cirrhosis (CC−) groups. CC+ patients were older, however, this difference was attributable to patients above the age of 70, where 28.15% were CC+ and 23.95% were CC− (*p* < 0.01). Compared to Caucasians (50.7% vs. 67.89%), there was a significantly higher distribution of Black (11.58% vs. 10.08%), Hispanic (28.17% vs. 15.45%), Asian or Pacific Islander (2.53% vs. 2.03%) and Native Americans (2.78% vs. 1.75%) in the CC+ versus CC− group (*p* < 0.01). Patients in the CC+ group had a lower income (38.14% vs. 33.49% in the <49,999$ group, *p* < 0.001) and were more likely to be insured by Medicare, and were more concentrated in the Mid-Atlantic, West South Central and Mountain Hospital Divisions. There was no difference observed between hospital size and teaching status. There was a higher prevalence of hypertension (35.04% vs. 32.51%), Diabetes mellitus(50.14% vs. 39.5%), chronic kidney disease (28.92% vs. 26.77%), and obesity (24.83% vs. 19.6%) in the CC+ vs. CC− group (*p* < 0.001). However, chronic pulmonary disease(24.78% vs. 22.28%), and smoking (45.82% vs. 34.31%) had a higher prevalence in the CC− group (*p* < 0.001). CC+ patients had a significantly higher NACSELD-ACLF score (13.82% vs. 10.12% score of 1) and 11.55% vs. 5.21% score > 2) (*p* < 0.001). Complications related to cirrhosis and portal hypertension during the hospitalisation were more likely in the CC− group, with more patients presenting with hepatorenal syndrome (5.2% vs. 3.51%), portal vein thrombosis (2.99% vs. 1.7%), spontaneous bacterial peritonitis (3.18% vs. 1.94%) and variceal bleeding (5.39% vs. 2.73%); *p* < 0.001. (Table 1).

### 3.2. In-Hospital Outcomes 

Our study reports a high in-hospital mortality overall of 52,620 patients (6.3%). After adjusting for potential confounding variables, in-hospital mortality was significantly higher in the CC+ group, with risk over three times compared to those in the CC− group (18.6% vs. 5.96%, *p* < 0.001, 3.39 (95% CI 3.08–3.74)). While age groups 30–49 and 50–69 were overrepresented in the alcoholic CC+ subgroup, only 14.35% of patients above the age of 70 remained in the alcoholic CC+ subgroup, compared to 34.7% in the NASH CC+ age group (*p* < 0.002). Additionally, there were significantly higher rates of complications in the alcoholic CC+ subgroup, as reported in previous studies, the CC+ group had over twice the rates of mechanical ventilation compared to the CC− group (19.92% vs. 9.07%, adjusted OR 2.71 2.71 (95% CI 2.51–2.93). Additionally, CC+ group had a 1.7 times likelihood of receiving vasopressors (4.12% vs. 2.45%, *p* < 0.001, adjusted odds ratio 1.71, 95% CI 1.46–2.01). CC+ patients had a 2.75-day higher length of stay and$26,253 higher total hospitalization charges compared to CC− patients. 

Overall, more patients with CC+ status were discharged to skilled nursing facilities compared to CC− patients (25.84% vs. 17.75%, *p* < 0.001). (Table 2) (Figure 1). 

### 3.3. Subgroup Analysis

We subsequently analysed differences between patient characteristics and outcomes based on etiology of cirrhosis for CC+ patients. From 28,610 patients with cirrhosis and COVID-19, 9198 patients (32.15%) had alcoholic cirrhosis. 24.67% patients in the alcoholic CC+ group were females, compared to 48.35% in the NASH CC+ patients (*p* < 0.001). There were no significant differences between NACSELD-ACLF scores. However, the rate of complications was significantly higher in the alcoholic CC+ subgroup. Patients with alcoholic CC+ had a higher occurrence of portal hypertension (30.22% vs. 14.97%, *p* < 0.001), hyponatremia (33.7% vs. 23.11%, *p* < 0.001), hepatorenal syndrome (7.12% vs. 1.8%, *p* < 0.001), portal vein thrombosis (2.39% vs. 1.37%, *p* = 0.006), spontaneous bacterial peritonitis (3.75% vs. 1.08%, *p* < 0.001), and variceal bleeding (5% vs. 1.65%, *p* < 0.001)(Figure 2). There was a small but significant decrease in mechanical ventilation requirement (19.89% vs. 19.94%, *p* = 0.01; adjusted odds ratio 0.81(95% CI 0.69–0.95)) and higher in-hospital mortality in the alcoholic CC+ subgroup (19.87% vs. 18.09%, *p* = 0.001; adjusted odds ratio 1.44(95% CI 1.17–1.77)) (Table 3).

### 3.4. Mortality Predictors 

In the multivariate Cox regression analysis to predit mortality, patients over the age of 70 had a significantly increased risk of mortality, (HR 2.59 (1.23–5.46), *p* < 0.01). Additionally, NACSELD-ACLF score was incrementally associated with increased mortality, with a score of 1 associated a HR of 2.94 (2.45–3.53), and score ≥ 2 with a HR of 4.79 (4.03–5.69); *p* < 0.001.

Spontaneous bacterial peritonitis (SBP) was associated with a slightly increased hazard ratio of 1.45 (1.01–2.08), *p* = 0.04 and interestingly, portal hypertension had a decreased hazard ratio for mortality, HR 0.79 (0.67–0.94), *p* = 0.008(Table 4).

## 4. Discussion

Our study reports poor outcomes in patients with COVID-19 and alcoholic and NASH cirrhosis, especially in the early phases of the pandemic, with the wild type and alpha variant virus in circulation, and represents unvaccinated population. We report over 3 times increase in mortality in patients with alcoholic and NASH cirrhosis and COVID-19 who were admitted to the hospital compared to patients with alcoholic and NASH cirrhosis alone. This association was independent of age, race, sex, comorbidities, as well as NACSELD-ACLF score. Our data corroborates with outcomes reported by previous studies performed on cohorts and national representative databases [17,18]. Data from several prospective studies has showed that patients with advanced liver disease are at an increased risk of morbidity with COVID-19 infection, including higher risks of hospitalisations, mechanical ventilation and death [19]. 

Mortality reported in our study is lower than what has been reported in other studies around the same time point. We report an in-hospital mortality of 18.6% in patients with alcoholic and NASH cirrhosis who were positive for COVID-19, which is significantly lower than reports in the same time frame. Bajaj et al. reported no difference in mortality rates between patients with cirrhosis who were positive versus negative for COVID-19, with a mortality of 30% in patients with cirrhosis and COVID-19 [8]. Similarly, case fatality for patients with cirrhosis and COVID-19 in a large European registry was reported at 31.4% [20]. One of the possible reasons for this can be exclusion of patients with other etiologies of cirrhosis, including viral hepatitis and autoimmune causes of cirrhosis. Additionally, patients who were admitted with COVID-19 were less likely to have complications associated with decompensated cirrhosis (such as portal hypertension, spontaneous bacterial peritonitis and variceal bleeding) than those without COVID-19, which corresponds to mortality rates reported by Marjot et al., in their Child-Pugh A subgroup analysis [21]. Additionally, the National Inpatient Sample database provides data for in-hospital mortality, and a significant portion (over 25%) were discharge to nursing facilities, and outcomes post discharge are unknown. It is also important to note that data from registries are susceptible to reporting bias, and may over-estimate mortality associated with COVID-19. Similar to reports of higher mortality in alcoholic liver disease reported by the COLD study group, as well as Marjot et al., we note an increased mortality in patients with alcoholic cirrhosis [15,21].This additional mortality risk persists after adjusting for comorbidities, therefore suggesting cirrhosis presents an increased risk in COVID-19, independent of its associated comorbidities. 

In line with previous studies [8,22], we describe a large proportion of patients with cirrhosis and COVID-19 requiring mechanical ventilation. While lung injury is the major established cause of morbidity and mortality in patients with COVID-19 [23], there is an implication of liver dysfunction as a potential driver of lung injury, with potential mechanisms including increased venous thromboembolic disease, altered pulmonary dynamics as well cirrhosis associated lung disease. Over expression of ACE2 enzyme in patients with cirrhosis may make them particularly vulnerable to the effects of COVID-19 [24]. We found that patients with COVID-19 and cirrhosis were less likely to have complications associated with decompensated liver disease, including portal vein thrombosis. Several reasons for this may exist, first, patients might have been primarily admitted for hypoxic respiratory failure from COVID-19 rather than decompensation of cirrhosis. Absence of admitting diagnosis might make this challenging. Second, patients with cirrhosis without COVID-19 were more likely to get admitted for complications of decompensated cirrhosis. As mentioned above, this may be a factor influencing the lower mortality rates reported in the study. 

Similar to results reported by Mackey et al. [25] and Gu et al. [26], racial minorities are over-represented in the COVID positive cirrhotic group. Hispanics seem to be at a particular disadvantage, with almost twice the number of COVID positive cirrhotic patients compared to COVID negative cirrhotic patients. As there is no evidence of increased susceptibility to COVID-19 in these groups, research is required to identify disparities in health access and exposure related factors, e.g., increased population density as well as delay in seeking care. CC+ patients with a low median household income had analogous increase in hospitalisation. Similarly, hospitalisations for CC+ were higher in patients above 70 years of age, as well as patients with comorbidities. This is consistent with current literature reports [13,20,27]. 

Patients with alcoholic cirrhosis and COVID were more likely to have complications associated with cirrhosis as well as increased mortality, compared to NASH cirrhosis. Active consumption of alcohol has previously shown to been a poor prognostic marker for alcoholic liver disease (ALD) associated cirrhosis [28]. In addition, ALD related cirrhosis has been shown to be associated with higher mortality due to decompensation events. Several reasons for this may exist- patients with alcoholic liver disease are reported to be sicker on presentation [28], heavy alcohol use may worsen prognosis of infectious complications due to impaired T lymphocyte cell function [29]. While there have been no studies reporting possible mechanisms of poor outcomes associated with COVID-19 and alcoholic cirrhosis, several studies have previously reported a high mortality in patients with alcoholic cirrhosis/liver disease and COVID-19 [9,15,30]. In addition, Shaheen et al. report a significant increase in alcoholic hepatitis or cirrhosis admissions compared to non-alcoholic cirrhosis during the COVID-19 pandemic [31]. 

Age greater than 70 years as well as a high NACSELD-ACLF score are significantly associated with an increase in mortality in our cohort. Our data also suggest that patients with an increasing number of organ failures have a high mortality with COVID-19, similar to what was reported in several other studies [22,32]. It is difficult to determine whether these organ failures resulted from inflammatory responses due to COVID-19 or due to worsening liver disease. 

Our study has several strengths. We use a nationally representative, gender balanced and diverse database that has been historically used to report healthcare and disease outcomes to perform our analysis. Due to temporally defined nature of our data, we are able to report outcomes in the pre-vaccination stage of the pandemic, with the wild type and alpha variants of the virus. 

We do recognise the limitations associated with the use of administrative data to define and delineate cirrhosis as well a possibility of attenuation of differences between infected and uninfected individuals as the analysis is limited to hospitalised patients. The absence of information during the admitting diagnosis may also make it challenging to ensure our study captures the entire data related to our patients of interest. 

## 5. Conclusions

Our study is one of the largest evaluating the outcomes of COVID-19 in patients with cirrhosis, and show an increased mortality and need for mechanical ventilation in patients with cirrhosis and COVID-19, prior to the administration of COVID-19 vaccination. Our study adds to the impetus of developing ways to mitigate the effect of COVID-19 in cirrhosis, such as pre-exposure prophylaxis for at-risk individuals as well as encouraging uptake of vaccination in the community. 

## Figures and Tables

**Figure 1 tropicalmed-07-00421-f001:**
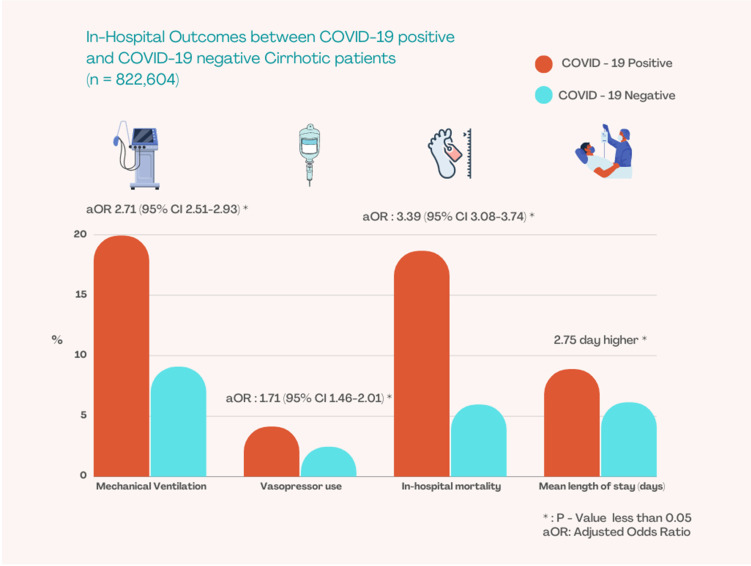
In-hospital Outcomes between COVID-19 positive and COVID-19 negative Cirrhotic patients.

**Figure 2 tropicalmed-07-00421-f002:**
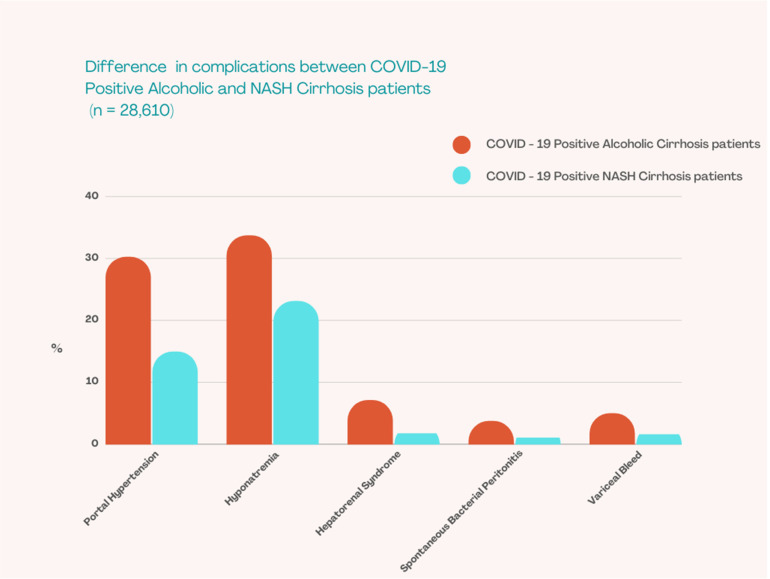
Difference in complications between COVID-19 positive alcoholic and NASH Cirrhosis patients.

**Table 1 tropicalmed-07-00421-t001:** Cirrhosis and COVID-19 patient-level characteristics.

Characteristics	COVID + Cirrhosis	COVID-Cirrhosis	*p* Value
**N = 822,604**	**N= 28,610 (3.47%)**	**N = 793,994 (96.53%)**	
**SEX (Female)**	40.74%	41.25%	0.47
**Mean age years (SD)**			<0.001
Male	60.51 (12.88)	59.64 (12.6)	
Female	63.18 (13.4)	61.03 (13.8)	
**Age Groups**			<0.001
≥18–29	321 (1.12%)	10,242 (1.29%)	
30–49	4629 (16.18%)	146,571 (18.46%)	
50–69	15,604 (54.54%)	447,019 (56.3%)	
≥70	8054 (28.15%)	190,162 (23.95%)	
**Race**			<0.001
Caucasians	14,505 (50.7%)	539,043 (67.89%)	
African American	3313 (11.58%)	80,035 (10.08%)	
Hispanics	8059 (28.17%)	122,672 (15.45%)	
Asian or Pacific Islander	724 (2.53%)	16,118 (2.03%)	
Native American	796 (2.78%)	13,895 (1.75%)	
Others	1216 (4.25%)	22,232 (2.8%)	
**Median Household Income**			<0.001
<49,999$	10,912 (38.14%)	265,909 (33.49%)	
50,000–64,999$	7482 (26.15%)	222,556 (28.03%)	
65,000–85,999$	5885 (20.57%)	177,061 (22.3%)	
>86,000$	4332 (15.14%)	128,468 (16.18%)	
Insurance Status			<0.001
Medicare	14,602 (51.04%)	385,484 (48.55%)	
Medicaid	6978 (24.39%)	205,168 (25.84%)	
Private	5791 (20.24%)	159,116 (20.04%)	
Self-pay	1239 (4.33%)	442,256 (5.57%)	
**Hospital Division**			<0.001
New England	1416 (4.95%)	41,844 (5.27%)	
Middle Atlantic	3599 (12.58%)	88,133 (11.1%)	
East North Central	4160 (14.54%)	115,129 (14.5%)	
West North Central	1851 (6.47%)	48,672 (6.13%)	
South Atlantic	4575 (15.99%)	159,593 (20.1%)	
East South Central	1631 (5.7%)	56,056 (7.06%)	
West South Central	4680 (16.36%)	103,854 (13.08%)	
Mountain	2615 (9.14%)	57,485 (7.24%)	
Pacific	4085 (14.28%)	123,228 (15.52%)	
**Hospital Bedsize**			0.51
Small	5816 (20.33%)	164,039 (20.66%)	
Medium	8266 (28.89%)	222,318 (28%)	
Large	14,531 (50.79%)	407,716 (51.35%)	
**Hosptal Teaching Status**			0.02
Rural	1971 (6.89%)	58,835 (7.41%)	
Urban non-teaching	4701 (16.43%)	140,934 (17.75%)	
Urban teaching	21,941 (76.69%)	594,304 (74.85%)	
**Comorbidities**			
Coronary Artery Disease	5130 (17.93%)	148,396 (18.69%)	0.16
Congestive Heart Failure	7069 (24.71%)	197,863 (24.92%)	0.73
Hypertension	10,025 (35.04%)	258,127 (32.51%)	<0.001
Diabetes Mellitus	14,345 (50.14%)	313,628 (39.5%)	<0.001
Chronic kidney disease	8274 (28.92%)	212,552 (26.77%)	<0.001
Chronic pulmonary disease	6375 (22.28%)	196,752 (24.78%)	<0.001
Obesity	7104 (24.83%)	155,623 (19.6%)	<0.001
Smoking	9816 (34.31%)	363,808 (45.82%)	<0.001
**Nacseld-ACLF Score**			<0.001
0	21,349 (74.62%)	672,275 (84.67%)	
1	3954 (13.82%)	80,352 (10.12%)	
≥2	3304 (11.55%)	41,367 (5.21%)	
**Complications**			
Portal Hypertension	5685 (19.87%)	230,179 (28.99%)	<0.001
Hyponatremia	7584 (26.51%)	200,007 (25.19%)	0.03
Hepatorenal syndrome	1004 (3.51%)	41,288 (5.2%)	<0.001
Portal vein thrombosis	486 (1.7%)	23,740 (2.99%)	<0.001
SBP ^#^	555 (1.94%)	25,249 (3.18%)	<0.001
Variceal bleed	781 (2.73%)	42,796 (5.39%)	<0.001
Hepatic encephalopathy	46 (0.16%)	1985 (0.25%)	0.16

^#^ Spontaneous Bacterial Peritonitis.

**Table 2 tropicalmed-07-00421-t002:** In-hospital outcomes in COVID + and COVID—patients with cirrhosis.

Variable	COVID+	COVID−	*p* Value
**Disposition**			<0.001
Home/Routine	15,395 (53.81%)	455,356 (57.35%)	
SNF/LTAC/Nursing home ^2^	7393 (25.84%)	140,934 (17.75%)	
Home health	5033 (17.59%)	168,962 (21.28%)	
AMA ^3^	790 (2.76%)	28,743 (3.62%)	
**Mechanical Ventilation**	5699 (19.92%)	72,015 (9.07%)	
	Adjusted odds ratio ^1^ 2.71 (95% CI 2.51–2.93)		<0.001
**Vasopressor use**	1179 (4.12%)	19,453 (2.45%)	
	Adjusted odds ratio ^1^ 1.71 (95% CI 1.46–2.01)		<0.001
**In-hospital mortality (N = 52,620)**	5339 (18.66%)	47,322 (5.96%)	
	Adjusted odds ratio ^1^ 3.39 (95% CI 3.08–3.74)		<0.001
**Mean total hospitalization charge ($)**	107,915$	79,340$	
	Adjusted total charge ^1^ 26,253$ higher		<0.001
**Mean length of stay (days)**	8.88	6.14	
	Adjusted length of stay ^1^ 2.75 day higher		<0.001

^1^ Adjusted for age, race, sex, weekend admission, income level, insurance status, discharge quarter, hospital characteristics, Elixhauser comorbidities and NACSELD score. ^2^ SNF: Skilled Nursing Facility, LTAC: Long term Acute Care hospital. ^3^ AMA: Leaving Against Medical Advice.

**Table 3 tropicalmed-07-00421-t003:** Sub-group analysis of COVID + Alcoholic and NASH Cirrhosis Patients.

Variable	COVID+ Alc Cirrhosis	COVID+ Nash Cirrhosis	*p* Value
**N = 28,610**	9198 (32.15%)	19,306 (67.48%)	
**Sex (Female)**	7058 (24.67%)	13,833 (48.35%)	<0.001
**Age Groups**			<0.001
≥18–29	312 (1.09%)	323 (1.13%)	
30–49	7324 (25.6%)	3353 (11.72%)	
50–69	16,871 (58.97%)	15,005 (52.45%)	
≥70	4106 (14.35%)	9928 (34.7%)	
**Nacseld-ACLF Score**			0.055
0	21,086 (73.7%)	21,475 (75.06%)	
1	3811 (13.32%)	4023 (14.06%)	
≥2	3717 (12.99%)	3110 (10.87%)	
**Complications**			
Portal Hypertension	8646 (30.22%)	4283 (14.97%)	<0.001
Hyponatremia	9642 (33.7%)	6612 (23.11%)	<0.001
Hepatorenal syndrome	2037 (7.12%)	515 (1.8%)	<0.001
Portal vein thrombosis	684 (2.39%)	392 (1.37%)	0.006
SBP ^1^	1073 (3.75%)	309 (1.08%)	<0.001
Variceal bleed	1431 (5%)	472 (1.65%)	<0.001
Hepatic encephalopathy	63 (0.22%)	37 (0.13%)	0.42
**Disposition**			<0.001
Home/Routine	15,349 (53.65%)	15,418 (53.89%)	
SNF/LTAC/Nursing home ^2^	7542 (26.36%)	7324 (25.6%)	
Home health	4401 (15.38%)	5324 (18.61%)	
AMA ^3^	1319 (4.61%)	544 (1.9%)	
**Mechanical Ventilation**	5691 (19.89%)	5705 (19.94%)	
	Adjusted odds ratio 0.81 (95% CI 0.69–0.95)		0.01
**Vasopressor use**	1167 (4.08%)	1187 (4.15%)	
	Adjusted odds ratio 0.83 (95% CI 0.6–1.15)		0.26
**In-hospital mortality (N = 5335)**	5685 (19.87%)	5176 (18.09%)	
	Adjusted odds ratio 1.44 (95% CI 1.17–1.77)		0.001
**Mean total hospitalization charge ($)**	114,751$	104,671$	
	Adjusted total charge 9053$ lower		0.10
**Mean length of stay (days)**	9.31	8.67	
	Adjusted length of stay 0.28 day lower		0.36

^1^ SBP: Spontaneous Bacterial Peritonitis. ^2^ SNF: Skilled Nursing Facility, LTAC: Long term Acute Care Hospital. ^3^ AMA: Leaving Against Medical Advice.

**Table 4 tropicalmed-07-00421-t004:** Mortality predictors in COVID-19 positive liver cirrhosis (Multivariate analysis).

Variable	HR (95% CI LL-UL)	*p* Value
Age ≥ 70	2.59 (1.23–5.46)	0.01
Male sex	0.93 (0.82–1.06)	0.33
**Nacseld-ACLF score**		
Score = 1	2.94 (2.45–3.53)	<0.001
Score ≥ 2	4.79 (4.03–5.69)	<0.001
Hypertension	0.94 (0.79–1.13)	0.56
Obesity	0.89 (0.76–1.04)	0.15
Chronic Kidney Disease	1.14 (0.96–1.35)	0.11
Hepatorenal Syndrome	1.30 (0.99–1.69)	0.051
Portal Hypertension	0.79 (0.67–0.94)	0.008
Variceal bleed	1.19 (0.86–1.65)	0.28
SBP ^1^	1.45 (1.01–2.08)	0.04

^1^ SBP: Spontaneous Bacterial Peritonitis.

## Data Availability

Restrictions apply to the availability of these data. Data was obtained from the National Inpatient Sample database, US.

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
