# Peer review of "COVID-19 Alcoholic Cirrhosis and Non-Alcoholic Steatohepatitis Cirrhosis Outcomes among Hospitalized Patients in the United States: Insight from National Inpatient Sample Database"

_tropicalmed, 2022, doi:10.3390/tropicalmed7120421_

Round 1

Reviewer 1 Report

Well written paper. The data might be very different in the era of post-vaccination, evidence based treatment options and milder COVID traits. Also the Chirrotics with COVID-19 might be primarily admitted for COVID ARDS rather than acute decompensation from cirrhosis. Absence of information on the admitting diagnosis might make it challenging which needs to be mentioned in the limitations. Fig 1 mentions Absolute relative risk instead of Absolute odd ratio - please correct. Please correct conclusion as mentioned in the pdf. 

Author Response

Dear reviewer, thanks for your time to review our manuscript and valuable suggestions.  

Comment: The absence of information on the admitting diagnosis might make it challenging, which needs to be mentioned in the limitations.

Response: we have added the statement in the limitations section. 

Comment:  Fig 1 mentions Absolute relative risk instead of Absolute odd ratio - please correct. 

Response: thanks for bringing this to our attention. It has been corrected. 

Comment: Please correct conclusion as mentioned in the pdf.

Response: We appreciate valuable suggestion from reviewer. we have made corrected the conclusion as suggested by reviewer in the PDF 

Sincerely 

RS

Reviewer 2 Report

I read it with great interest, but I have raised several concerns.

#1. Please add the hypothesis of your study in introduction section.

#2. Multivariate linear regression model was used for continuous variables (LOS and Total hospital charge) -> Please cite the statistical guideline such as DOI: https://doi.org/10.54724/lc.2022.e3

#3. The authors have to perform the PSM technique.

#4. Limitations are too short to draw your findings.

#5. This is an excellent paper.

Author Response

We would like to thank the reviewers for their time to review this manuscript, and we appreciate their feedback. Please find our responses below. Edits have been made in the manuscript that can be found in the track changes section.

#1. Please add the hypothesis of your study in introduction section.

We thank the reviewer for this suggestion, and have included our hypothesis in the introduction section, page2, lines 66-68…” We hypothesize significant morbidity and mortality in patients with cirrhosis and COVID-19 infections, with respiratory compromise as the driving force for decompensatory episiodes.”

#2. Multivariate linear regression model was used for continuous variables (LOS and Total hospital charge) -> Please cite the statistical guideline such as DOI: https://doi.org/10.54724/lc.2022.e3

We thank the reviewer for this valuable comment, and have included this citation in our methods section, on page 3, line 114

#3. The authors have to perform the PSM technique.

We appreciate this suggestion from the reviewer, however, would like to bring forth that while propensity score matching, an enormously popular method of preprocessing data for causal inference may often increase imbalance, model dependency and bias. In weighted data samples such the National Inpatient Sample database with an average incidence of the studied variable, data is typically balanced enough to approximate complete randomization. (King et al, King, G., & Nielsen, R. (2019). Why Propensity Scores Should Not Be Used for Matching. Political Analysis, 27(4), 435-454. doi:10.1017/pan.2019.11)

#4. Limitations are too short to draw your findings.

We thank the reviewer’s comments, and have expanded our limitations, which now read as … “The absence of information during the admitting diagnosis may also make it challenging to ensure our study captures the entire data related to our patients of interest...” page 10 line 280-283

#5. This is an excellent paper.

We thank the reviewer for the encouraging worlds 

Sincerely 

RS 

Round 2

Reviewer 2 Report

This is an excellent paper.